# TMJ Position in Symmetric Dentofacial Deformity

**DOI:** 10.3390/jcm11133631

**Published:** 2022-06-23

**Authors:** Victor Ravelo, Gabriela Olate, Marcio de Moraes, Henry Garcia Guevara, Marcelo Parra, Sergio Olate

**Affiliations:** 1Research Undergraduate Group in Dentistry (GIPO), Faculty of Health Sciences, Universidad Autónoma de Chile, Temuco 4810101, Chile; victor.ravelo.s@gmail.com; 2Center of Excellence in Morphological and Surgical Studies (CEMyQ), Universidad de La Frontera, Temuco 4811230, Chile; gaby.olate@gmail.com (G.O.); marcelo.parra@ufrontera.cl (M.P.); 3Division of Oral and Maxillofacial Surgery, Piracicaba Dental School, State University of Campinas, Piracicaba 13400-001, Brazil; marciom@unicamp.br; 4Division of Oral and Maxillofacial Surgery, Hospital Ortopedico Infantil, Caracas 1050, Venezuela; henryagg@gmail.com; 5Department of Oral Surgery, Santa Maria University, Caracas 1073, Venezuela; 6Division of Oral, Facial and Maxillofacial Surgery, Dental School, Universidad de La Frontera, Temuco 4811230, Chile

**Keywords:** temporomandibular joint, symmetry, orthognathic surgery, dentofacial deformity

## Abstract

The aim of this research was to analyze the facial class, presence of malocclusion, and the mandibular plane and to relate this to the mandibular condyle position. A cross-sectional study in subjects under analysis for orthognathic surgery was done. The mandibular plane, the gonial angle, and the molar class were included to compare the coronal and sagittal position of the condyle and the joint space observed in the CBCT. The measurements were obtained by the same observer at an interval of two weeks. In addition, the Spearman test was performed to determine the correlation using a *p* value < 0.05 to observe any significant differences. Eighty-nine male and female subjects (18 to 58 years old, 24.6 ± 10.5) were included. In the coronal section, subjects with CIII had a greater mediolateral distance (MLD, *p* = 0.0001) and greater vertical distance (SID, *p* = 0.0001) than subjects with CII. In terms of the skeletal class and the mandibular plane, it was observed that subjects in the CII group had a greater mandibular angle (open angle) (*p* = 0.04) than the CII group and was related to the anterior position of the condyle. The most anterior condylar position was observed in the CII group (*p* = 0.03), whereas a posterior condylar position was significant in CIII subjects (*p* = 0.03). We can conclude that the sagittal position of the TMJ was related to the mandibular plane and the skeletal class showing a higher mandibular angle and most anterior position of the condyle in CII subjects and a lower mandibular angle and most posterior position of the condyle in CIII subjects. The implications for surgical treatment have to be considered.

## 1. Introduction

The temporomandibular joint (TMJ) includes some anatomical structures; clinical variables could affect or modify its morphology and position due to remodeling or changes on the surface as an adaptive response [1]. This adaptation could be an anatomical variation, functional response, or pathological condition [2].

Morphological changes in the TMJ and facial morphology can be related [3] because the condylar volume is related to the size of the mandibular neck, ramus, or body—mainly in the growing stage [4]. In fact, facial asymmetry is strongly connected to the volumetric condition of the condyle [5].

Some authors show that malocclusions and facial deformities have been related to the position and morphology of the TMJ by observing a shift in the mid-line, changes in mandibular movements, or facial pain [6]. On the other hand, other authors show that malocclusion is not related to pathological alterations of the TMJ [7,8,9,10,11].

Previous observations [12,13] indicate that the shape of the mandibular condyle and TMJ are related to facial morphology, and surgical treatment must include this variable.

The position of the condyle is an important finding for the diagnosis and planification of orthognathic surgery [13]. Any error in this analysis could lead to relapse or complications in the follow up [4]. In the 3D era, the direct analysis of the condylar position is necessary [1] because the tridimensional stability of the maxillomandibular movement will be influenced by TMJ position. In this sense, the size of the condyle could be related to the disk and ligaments and, finally, be related to the function of the TMJ [14].

Previous findings showed that the anteroposterior condylar position in subjects with non-surgical malocclusions showed differences between class II dental occlusions and normal dental occlusions but failed to find any differences in class III dental occlusions [15]. In subjects under surgical treatment for malocclusions, the morphology of the condyle could be different, but this is not a consensus in class II or class III dentofacial deformity.

The aim of this research was to analyze the facial class, presence of malocclusion, and the mandibular plane as they relate to the mandibular condyle position (Appendix A).

## 2. Materials and Methods

Eighty-nine subjects were included in this cross-sectional study that analyzed the angulation of the mandibular plane, dental occlusion, and position of the sagittal and coronal condyle in the temporo-mandibular joint (TMJ) in subjects with class II and class III dentofacial deformities. The subjects signed an informed consent form, and the study was conducted by protecting the participants’ integrity and respecting the Declaration of Helsinki. The research protocol was approved by the ethics committee of the Universidad de La Frontera (protocol: 056/21)

A 3D image obtained by cone beam computed tomography (CBCT) was used and analyzed in the NewTom 3D software, VGi EVO model (Verona, Italy) with visualization field of 24 cm × 19 cm and exposure parameters of 110 kV, 8 mA, 15 s. The image was obtained by a radiologist; the patient was placed in a vertical position, keeping the lips at rest, without forcing a body position.

Male and female subjects in the pre-operative stage of orthognathic surgery were included; all of them showed a bilateral upper and lower first molar or skeletal deformity class II (>4° in the ANB angle) or class III (<0° in the ANB angle). The facial symmetry of the participants was confirmed by clinical evaluation with a clinical deviation of the chin up to 5 mm, and dental symmetry was assessed by observing a clinical deviation of the dental midline up to 3 mm. We excluded subjects with an absence of key teeth, such as incisors, canines, bicuspids, or the first molar; previous facial surgery; history of facial trauma; presence of facial syndromes; or conditions with significant morphological changes.

### 2.1. Mandibular Plane (Angle)

To determine the angle of the mandibular plane, the McNamara analysis was used, which uses the intersection of the points Po-Or and Go-Ch (Go: Gonion, point located in the most posterior and lower area of the mandibular angle; Ch: Chin, lower point of the mandibular symphysis), with a normality parameter of 25° ± 4° (Figure 1).

### 2.2. Gonial Angle

To determine the gonion angulation, the Jarabak analysis was used, which takes the intersection of the points J-Go-Ch (J: joint, point located at the intersection of the posterior edge of the ramus and basilar process of the occipital bone; Ch: Chin, most inferior point of the mandibular symphysis), with a parameter of 130° ± 7° (Figure 1).

### 2.3. Joint Space (Sagittal Measurements)

To measure the position of the condyle in the TMJ, the anatomical references proposed by Vitral et al. [14] were used. The anterior joint space (AJS), upper joint space (UJS), and posterior joint space (PJS) were measured (Figure 2 and Figure 3).

Anterior joint space (AJS): The distance was measured between the most anterior point of the condylar surface and the posterior wall of the articular tubercle.

Upper joint space (UJS): The distance was measured between the upper point of the condylar surface and the most upper point of the articular fossa.

Posterior joint space (PJS): The distance was measured between the posterior point of the condylar surface and the posterior wall of the articular fossa.

### 2.4. Joint Space (JS) Average

The methodology proposed by Fraga et al. [15] was used to calculate the ratios of the anterior and posterior (A/P) space. It was calculated by dividing the anterior joint space by the posterior joint space, thereby obtaining 1 (±0.09) mm as the concentric condylar position.

### 2.5. Coronal Analysis of the Condyle

Using the methodology proposed by Muñoz et al. [16], a central image of the coronal plane was used to take measurements by means of the vertical and transverse orientations of the condyle (Figure 4).

Mediolateral distance of the condyle (MLD): The measurement was obtained from a transverse line of the condyle, which begins and ends at its most medial and most lateral cortical point.

Vertical distance of the condyle (VD): The measurement was obtained from a vertical line that begins at the highest cortical point and the lowest point of the condylar head.

### 2.6. Molar Class

To obtain a reference of the occlusal position, the classification proposed by Angle was used—which refers to the occlusal relation of the permanent first molar.

Class I occlusion (MCI): The mesiobuccal cusp of the maxillary first molar is aligned directly above the vestibular sulcus of the mandibular first molar.

Class II occlusion (MCII): The mesiobuccal cusp of the upper first molar is aligned above the vestibular sulcus of the mandibular first molar.

Class III occlusion (MCIII): The mesiobuccal cusp of the maxillary first molar is aligned posterior to the vestibular sulcus of the mandibular first molar.

The measurements were taken by the same observer at different times with a two-week difference between them by using 30 studies to calibrate. For the categorical variables, a kappa index = 0.75 was obtained, and for the continuous variables, an intraclass index of 0.73 was obtained. A 95% confidence interval was used to measure the agreement between the sagittal and coronal measurements of the TMJ, the molar occlusal relationship, and facial deformity in terms of the sagittal maxillomandibular relations. In addition, the Spearman test was performed to determine the correlation among the variables. A value of *p* < 0.05 was considered to determine any significant differences. A summary of measurements and landmarks is included in Table 1.

## 3. Results

Eighty-nine subjects were included with an age between 18 to 58 years old (24.6 ± 10.5). Thirty-six were male (40.4%) and fifty-three were female (59.5%). The male group showed a greater upper joint space and condylar measurement with no statistical differences (Table 2).

In terms of dentofacial deformity (FD), 56 subjects presented a skeletal type II FD (CII) and 33 presented a skeletal type III FD (CIII) (Table 3). CII subjects showed greater distances in the joint space (SJS, *p* = 0.42 and EAP, *p* = 0.12) with no statistical differences. However, the size of the mandibular condyle presented significant differences, with CIII subjects having a greater mediolateral distance (MLD, *p* = 0.0001) and upper-to-lower distance (VD, *p* = 0.0001) than CII subjects.

Dental occlusion was related to facial deformity (*p* = 0.0001) (Table 2); CII subjects showed a significantly higher mandibular angle (*p* = 0.04) than CIII subjects. In the same direction, subjects with CII presented a greater gonial angle (*p* = 0.03) than subjects with CIII, and the mandibular condylar position was strongly related to the mandibular plane (*p* = 0.03). The differences between CII and CIII subjects were observed in the anterior condylar position in the glenoid fossa; CII subjects showed a more anterior position than CIII subjects (Table 4).

## 4. Discussion

CBCT is a highly accurate three-dimensional method for the diagnosis of bone structures of the TMJ, as well as changes in condylar morphology [17,18].

In this research, CBCT was used as a diagnostic tool for face analysis. We found that the condyle and its position within the articular fossa were related to the angle of the mandibular plane and the skeletal class, where subjects with the highest mandibular plane presented a more anterior condylar position. Saccucci et al. [19] used a comparable sample that showed that the condylar volume and facial high could affect the morphology of the TMJ, where the subjects with a smaller mandibular plane showed a greater mandibular plane and an anterior condylar position.

Katsavarias et al. [20] concluded that the subjects with skeletal class II had an anterior positioning of the condyle and the subjects with skeletal class III had a posterior positioning. In our sample, the results were in the same direction because the subjects with skeletal class III showed a lower mandibular plane and a posterior condylar position. Some authors [21,22,23,24] observed a relation between the skeletal class and the sagittal and coronal position of the condyle, indicating that this morphology could explain the differences in the adaptive biomechanical patterns of the TMJ for different facial morphologies.

Our findings support the statement that the TMJ is related to facial morphology. Goulart et al. [25] indicated that the condylar volume in subjects with mandibular prognathism was similar to subjects with unilateral condylar hyperplasia, concluding that the facial morphology in subjects with skeletal class III could be related to the presence of a bilateral condylar hyperplasia [26]. Likewise, our results show that the coronal distance of the condyle is significantly greater in subjects with skeletal class III, thereby supporting the previous conclusion of Goulart et al. [25,26].

Rodríguez et al. [27] evaluated the TMJ morphology and position of 30 subjects with a class I dental occlusion and no differences were noted in the symmetry or the sagittal position of the condyles. Merigue et al. [28], with a sample of 49 subjects with class I and class II dental occlusions, failed in to find any relation between the condylar position and the molar occlusion. In our research, dental occlusion was related to facial morphology; the mandibular plane and gonial angle showed a strong relation with the condylar size and condylar position in the glenoid fossa, supporting the relation between dental occlusion, facial morphology, condylar size, and condylar position.

Some authors showed no relation between dental occlusion and TMJ pathology [8,29,30]; however, facial or skeletal conditions could show some relation with the development or natural evolution of some TMJ pathologies. In this research, no TMJ pathology was diagnosed, and all the subjects were involved in orthognathic surgery due to facial deformity, the related dental occlusion, and the deficiencies in maxillofacial function, which were included in a full diagnosis to indicate the need for facial surgery. The value of the mandibular condyle in facial growth has been confirmed [31], which implies that the TMJ is a very important key point in the maxillomandibular relation and, consequently, in the position of the condyle and the maxillofacial function.

Orthognathic surgery in patients with facial deformity has an impact on the TMJ morphology and position, as well as on the masticatory muscles and surrounding soft tissues [32,33].

The health or disease of the TMJ must be evaluated before making a surgical plan in order to involve these findings in the surgery. In this sense, a systematic review concluded that orthognathic surgery could bring about changes in the condylar position and reduce the symptoms of TMJ disorders [34,35,36]; on the other hand, Mohlhenrich et al. [37] observed that the condylar position after surgery is also influenced by the type of movement and the surgical technique, so a diversity of variables could induce the final condition and function of the TMJ.

The limitations of this research are related to the absent use of magnetic nuclear resonance, given that it is a gold standard in TMJ evaluation; however, this is not a regular exam for orthognathic surgery.

## 5. Conclusions

The sagittal position of the mandibular condyle was related to the mandibular plane and the maxillomandibular relations; the subjects with skeletal class II showed a greater mandibular plane, which was also associated with the anterior position of the condyle. Subjects with class III presented condyles with a larger diameter than subjects with class II.

## Figures and Tables

**Figure 1 jcm-11-03631-f001:**
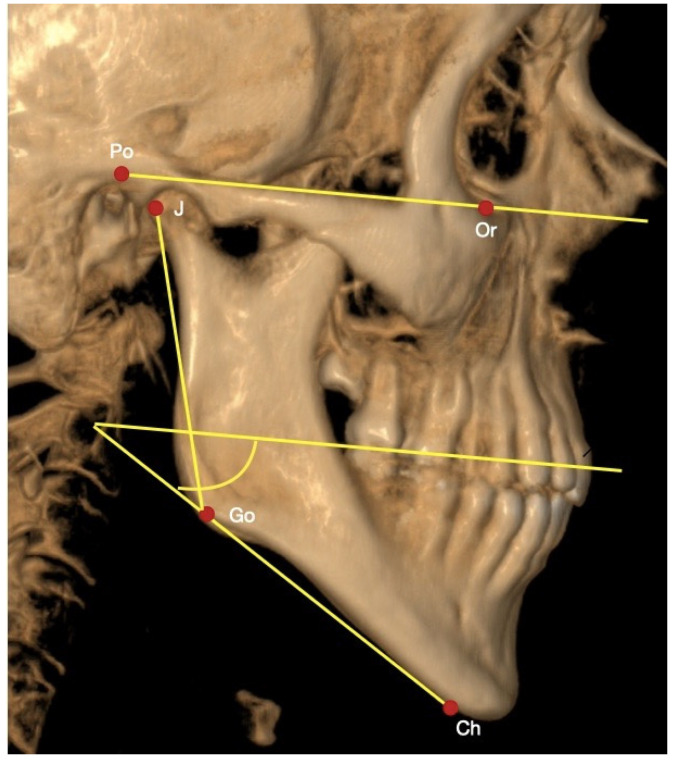
CBCT with landmarks used in the analysis. Ch: chin, Go: gonion, Co: condylion, Po: porion, Or: orbital; J: joint.

**Figure 2 jcm-11-03631-f002:**
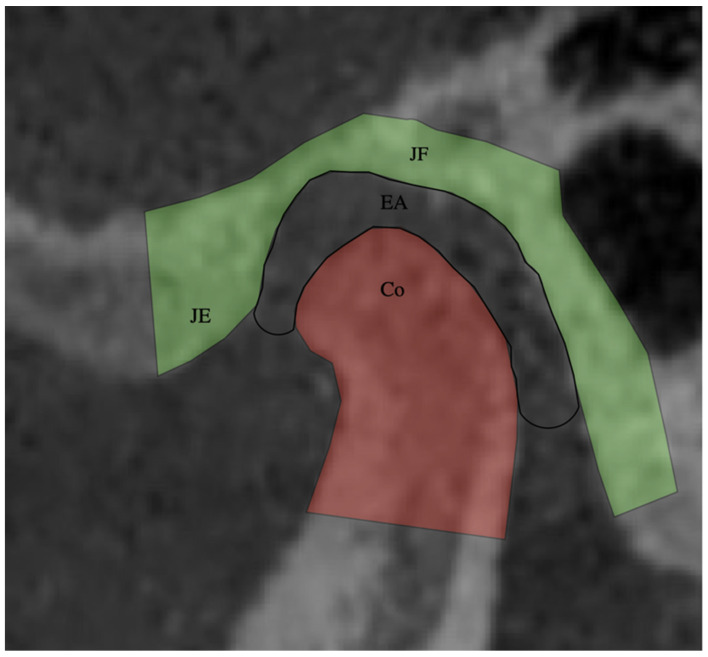
TMJ in sagittal view. JE: articular eminence; JF: glenoid fossa; EA: joint space; Co: condylar head.

**Figure 3 jcm-11-03631-f003:**
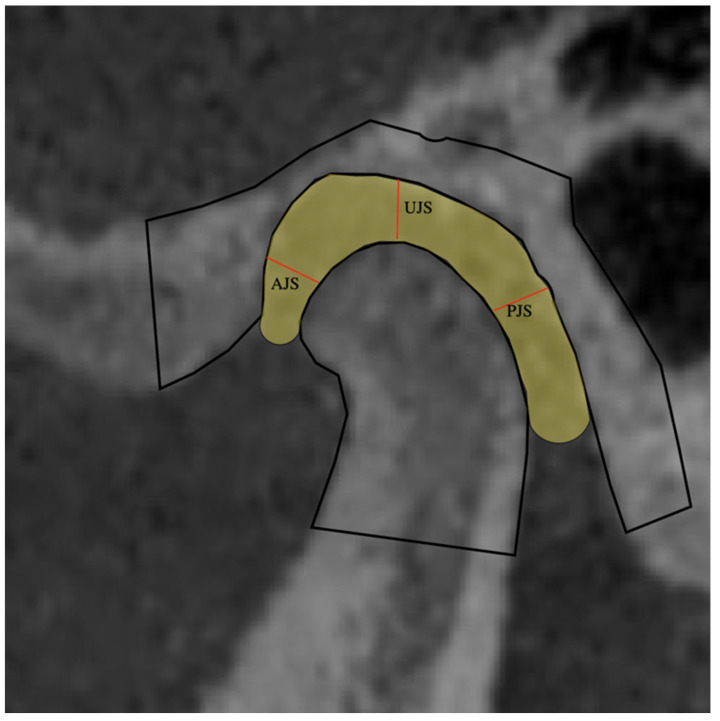
Joint space in the upper level of the condyle. AJS: Anterior joint space; UJS: upper joint space; PJS: posterior joint space.

**Figure 4 jcm-11-03631-f004:**
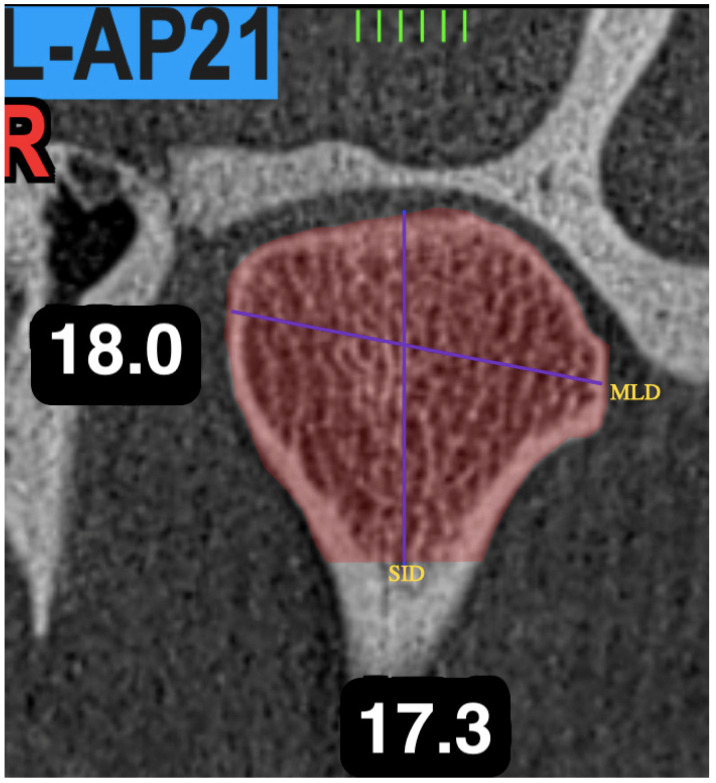
Coronal view of the TMJ. MLD: mid-lateral distance; SLD: upper-lower distance.

**Table 1 jcm-11-03631-t001:** Landmark and measurement used in this research for skeletal and condylar analysis.

Item	Measurement Strategy
Mandibular plane	Po-Or/Go-Ch (Normality: 25° ± 4)
Gonial angle	J-Go-Ch (Normality: 130° ± 7)
Anterior Joint Space	Distance between the most anterior point of the condylar surface and the posterior wall of the articular tubercle.
Upper joint Space	Distance between the upper point of the condylar surface and the most upper point of the articular fossa.
Posterior Joint Space	Distance between the posterior point of the condylar surface and the posterior wall of the articular fossa.
Mediolateral Distance of the Condyle	Distance between the most medial and most lateral cortical point of the condyle.
Vertical Distance of the Condyle	Distance between the highest cortical point and the lowest point of the condylar head

Note: Ch: chin, Go: gonion, Co: condylion, Po: porion, Or: orbital; J: joint (point located at the intersection of the posterior edge of the ramus and basilar process of the occipital bone).

**Table 2 jcm-11-03631-t002:** Distribution of subjects according to gender and the sagittal and coronal morphology of the TMJ.

	Male (*n*: 36)				Female (*n*: 53)				
	Right TMJ		Left TMJ		Right TMJ		Left TMJ		
	X	SD	X	SD	X	SD	X	SD	*p* Value
AJS	2.21 mm	0.88	2.32 mm	4.74	2.42 mm	0.94	2.43 mm	0.93	0.12
UJS	2.75 mm	1.15	2.67 mm	1.03	2.30 mm	0.96	2.42 mm	0.87	0.25
PJS	2.34 mm	1.34	2.26 mm	1.25	2.55 mm	1.40	2.58 mm	1.07	0.41
MLD	21.90 mm	4.67	22.07 mm	4.75	19.99 mm	3.40	19.82 mm	3.74	0.009 *
VD	19.37 mm	4.49	20.12 mm	4.23	18.52 mm	4.22	18.57 mm	4.01	0.47

Note: AJS: anterior joint space. UJS: upper joint space. PJS: posterior joint space. MLD: med-lateral distance. VD: vertical distance. X: average measurement; SD: standard deviation. (*) indicate significant difference.

**Table 3 jcm-11-03631-t003:** Distribution of subjects according to skeletal class and the sagittal and coronal morphology of the TMJ.

	Facial CII (*n*: 56)				Facial CIII (*n*: 33)				
	Right TMJ		Left TMJ		Right TMJ		Left TMJ		
	X	DS	X	DS	X	DS	X	DS	*p* Value
AJS	2.34 mm	0.86	2.49 mm	0.82	2.33 mm	1.03	2.43 mm	1.05	0.32
UJS	2.53 mm	1.14	2.61 mm	0.98	2.35 mm	0.89	2.37 mm	0.86	0.42
PJS	2.73 mm	1.48	2.58 mm	1.26	1.97 mm	1.02	2.22 mm	0.90	0.12
MLD	19.43 mm	3.90	19.66 mm	4.41	23.03 mm	3.24	22.55 mm	3.48	0.0001 *
VD	17.07 mm	4.05	17.98 mm	4.19	21.91 mm	2.86	21.27 mm	3.16	0.0001 *

Note: CII: skeletal class II. CIII: skeletal class III. AJS: anterior joint space. UJS: upper joint space. PJS: posterior joint space. MLD: mediolateral distance. VD: vertical distance. X: average measurement; SD: standard deviation. (*) indicates significant difference.

**Table 4 jcm-11-03631-t004:** Distribution of subjects according to skeletal class, mandibular plane, and the sagittal and coronal position of the TMJ.

		CII (*n*: 56)		CIII (*n*: 33)		
		*n*	%	*n*	%	*p* Value
Mandibular Plane	Low	2	2.24	31	34.83	
	High	43	43.87	12	13.48	0.04 *
Condylar position	Centric	12	12.24	6	6.12	
	Anterior	27	27.55	8	8.16	
	Posterior	17	17.34	19	19.38	0.03 *

Note: CII: skeletal class II. CIII: skeletal class III; Mn: mandible; *n*: number of subjects. (*) indicates significant difference.

## Data Availability

Data are available upon reasonable request.

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
