# Peer review of "TMJ Position in Symmetric Dentofacial Deformity"

_jcm, 2022, doi:10.3390/jcm11133631_

Round 1
Reviewer 1 Report
Dear authors, I read with interest your article. I consider that the topic is of interest to both orthodontists and oral and maxilo-facial surgeons. Bellow, you can find my recommendations.
Abstract - I would recommend revising it to be more accurate
Keywords - I recommend removing either the word TMJ or the temporomandibular joint because they are the same
Introduction section
You can discuss and emphasize more the interaction between facial deformities and the TMJ.
"Morphological changes in the TMJ and the facial morphology can show a relation [3] because the condylar volume is related to the size of mandibular neck, ramus or body, mainly in the growing stage[4]". - here is missing a space.
The Aim of the study written in this section is different from the one in the Abstract.
Materials and Methods section
I would suggest introducing in this section a phrase with the number of subjects included in the study.
I kindly advise you to specify what type of symmetry are you referring to in the following phrase (facial, dental....) "symmetry of participants was confirmed by the clinical evaluation with a clinical deviation of the chin up to 5 mm and clinical deviation of the dental midline up to 3 mm".
"Were excluded subjects with absence of tooth......." - Please be more precise and specify which tooth or how many teeth...
"Ch: Chin, most inferior point of the mandibular symphysis), with a normality parameter of 130º ± 7"
"Class I occlusion (MCI):" - please remove the underline from this.
Introduce the figures numbers in the text, because they are attached but not mentioned in the text.
If it is possible, I would suggest explaining what you mean by facial deformity (FD), because is not very clear from the text - it is an abnormality of the face, an abnormality of the occlusion, or something else?
Results section
"The male group showed greater superior joint space and condylar measurement with no statistical differences (Table I)." - I think there should be "upper" instead of "superior".
"Dental occlusion was related to facial deformity (p=0.0001)" - I recommend at the end of the phrase to introduce the reference to the table - Table II.
Consider rewriting this "Differences between CII and CIII subjects were observed in the anterior condylar position in09oi80 the glenoid fossa showing CII subjects with a more anterior position than CIII subjects (Table III)."
Discussion section
Consider the revision of these phrases.
"The condition of the TMJ is evaluated previously to surgical plan to involve these findings into the surgery because in a systematic review was concluded that orthognathic surgery can bring about changes in condylar position and to get a reduction in the symptoms of TMJ disorders [34,35,36]; Mohlhenrich et al. [37] observed that this change is also influenced by the type of movement and the surgical technique. Limitations of this research are related to the absent use of magnetic nuclear resonance as a gold standard in TMJ evaluation, however, this is not a regular exam in orthognathic surgery."
Conclusion
I would recommend rephrasing/reformulating the conclusion. You make here refers to the "facial condition" and is not clear to me which are these facial conditions.
Reviewer 2 Report
The manuscript title."Mandibular Condyle Position in Non Asymmetric Facial Deformity" The aim of this research was to evaluate the position of the TMJ in subjects with class II 13 (CII) and class III (CIII) non asymmetric facial deformities.
-The title includes the condyle position only while the research analyse ( the angulation of the mandibular plane, dental occlusion, position of the sagittal and coronal condyle in the temporo-mandibular joint (TMJ) in subjects with class II and class III facial deformities.) Please revise.
- "non Asymmetric" would you confirm this term?
-The second aim was to analyze the facial class, presence of malocclusion, and the mandibular plane and relate to the mandibular condyle position.
-The introduction needs more input to explain the current finding related to the topic and what is the main problem.
Materials and Methods:
- I would like to see a schematic or CBCT showing all analyzed items, which you may already explained in words or some of them in figures:
- Mandibular plane (angle)
- Gonial angle
- Joint space (sagittal measurements)
- Joint space (JS) average
- Coronal analysis of the condyle
- Molar class
Author Response
Reviewer. The manuscript title. "Mandibular Condyle Position in Non Asymmetric Facial Deformity" The aim of this research was to evaluate the position of the TMJ in subjects with class II 13 (CII) and class III (CIII) non asymmetric facial deformities.
-The title includes the condyle position only while the research analyse ( the angulation of the mandibular plane, dental occlusion, position of the sagittal and coronal condyle in the temporo-mandibular joint (TMJ) in subjects with class II and class III facial deformities.) Please revise.
- "non Asymmetric" would you confirm this term?
Reply: Dear reviewer, thank for your comment. The title was moved to: “TMJ Position in Symmetric Dentofacial Deformity”
Reviewer. The second aim was to analyze the facial class, presence of malocclusion, and the mandibular plane and relate to the mandibular condyle position.
Reply: The aim was modified in the abstract and the introduction to maintain the same writing.
Reviewer. The introduction needs more input to explain the current finding related to the topic and what is the main problem.
Reply: a new paragraph was added in the introduction to be more accuracy in the main problem.
Reviewer. Materials and Methods:
- I would like to see a schematic or CBCT showing all analyzed items, which you may already explained in words or some of them in figures:
- Mandibular plane (angle)
- Gonial angle
- Joint space (sagittal measurements)
- Joint space (JS) average
- Coronal analysis of the condyle
- Molar class
Reply: was included a new figure (Figure 1) showing the measurement and the Table I to describe the measurements in the same format.
Round 2
Reviewer 1 Report
Dear authors,
I read the revisions that you made so far. There are some minor recommendations concerning text editing.
Abstract - A cross-sectional study in subjects under analysis for orthognathic surgery were realized - should be was realized or was conducted.
The measurements were obtained by the same observer twice at two-week - in order to be more clear I would suggest writing "at an interval of two-week".
Keywords: I would suggest replacing facial deformity with dentofacial deformity and replacing ";" after the last keyword with "."
Materials and Methods
I would suggest the following:
- In the phrase "facial symmetry of participants were confirmed by the clinical evaluation with a clinical deviation of the chin up to 5 mm and dental symmetry was assessment observing a clinical deviation of the dental midline up to 3 mm" I would suggest replacing were confirmed with was confirmed and "was assessment observing" with was assessed observing
- removing the definition of the J point that is written between the parenthesis (point located at the intersection of the posterior edge of the ramus and basilar process of the occipital bone) from the title of Figure 1: CBCT with landmarks used in the analysis, because the definition is already mentioned in the text.
- replacing the word "join" with "joint" in Figure 3
